# Synergistic Effect of Ribitol and Shikonin Promotes Apoptosis in Breast Cancer Cells

**DOI:** 10.3390/ijms26062661

**Published:** 2025-03-15

**Authors:** Ravi Doddapaneni, Jason D. Tucker, Pei J. Lu, Qi L. Lu

**Affiliations:** McColl-Lockwood Laboratory for Muscular Dystrophy Research, Cannon Research Center, Carolinas Medical Center, Atrium Health, Charlotte, NC 28203, USA

**Keywords:** breast cancer, chemotherapy, shikonin, ribitol, metabolomics, drug synergy

## Abstract

The mortality rate of breast cancer remains high, despite remarkable advances in chemotherapy. Therefore, it is imperative to identify new treatment options. In the present study, we investigated whether the metabolite ribitol enhances the cytotoxic effect of shikonin against breast cancer in vitro. Here, we screened a panel of small molecules targeting energy metabolism against breast cancer. The results of the study revealed that ribitol enhances shikonin’s growth-inhibitory effects, with significant synergy. A significant (*p* < 0.01) increase in the percentage (56%) of apoptotic cells was detected in the combined treatment group, compared to shikonin single-treatment group (38%), respectively. The combined ribitol and shikonin treatment led to significant arrest of cell proliferation (40%) (*p* < 0.01) compared to untreated cells, as well as the induction of apoptosis. This was associated with upregulation of p53 (*p* < 0.05) and downregulation of c-Myc (*p* < 0.01), Bcl-xL (*p* < 0.001), and Mcl-1 (*p* < 0.05). Metabolomic analysis supports the premise that inhibition of the Warburg effect is involved in shikonin-induced cell death, which is likely further enhanced by dysregulation of glycolysis and the tricarboxylic acid (TCA) cycle, afflicted by ribitol treatment. In conclusion, the present study demonstrates that the metabolite ribitol selectively enhances the cytotoxic effect mediated by shikonin against breast cancer in vitro.

## 1. Introduction

Recent research has primarily focused on investigating the potential of anti-cancer compounds to target specific molecular pathways involved in cancer progression [1,2,3,4,5]. These compounds have shown promising results in inhibiting tumor growth, reducing metastasis, and enhancing the sensitivity of cancer cells to chemotherapy. These therapeutic strategies show an enhanced anti-tumor effect, resulting in a notable improvement in survival rates. However, significant challenges persist in the field of drug therapy, including multi-drug resistance, poor response in certain breast cancer types, and undesirable dose-related side effects [6,7,8,9,10,11]. Therefore, the identification of agents that address these limitations and achieve synergistic effects with reduced doses of existing drugs becomes a desirable solution. Cancer cells are known to adapt their metabolism, such as their aerobic glycolysis (Warburg effect), to provide resistance to drugs, as well as to encourage cell survival and proliferation [12,13,14,15,16]. Thus, modulation of metabolism offers great potential to counter drug resistance and enhance efficacy.

We previously reported that the pentose alcohol ribitol had the capacity to enhance the production of CDP-ribitol and restore matriglycan expression in an LGMD dystroglycanopathy mouse model with FKRP mutations, leading to significant improvement in muscle pathology and function [17,18]. Ribitol is a metabolite found in nature, and has already been tested for daily use in animal models and clinical trials in humans without severe side effects, providing a safety profile for it to be trialed in combination with any drug for different treatment purposes. We also demonstrated that ribitol was able to enhance matriglycan in breast cancer cells, which could minimize invasion and metastasis [19].

More recently, our studies on untargeted global metabolomics demonstrated that ribitol supplementation affects a wide range of metabolic pathways with a distinctive profile in glycolysis, including the pentose phosphate pathway (PPP) and the tricarboxylic acid (TCA) cycle [20]. To explore the potential of ribitol-induced metabolic alterations for enhancing anti-cancer drug effects, we previously investigated the combinatorial effects of ribitol with anti-cancer drugs in breast cancer cells. We reported that the combination of ribitol with JQ1 synergistically inhibited the proliferation and migration of triple-negative MDA-MB-231 breast cancer cells (TNBC), but not of MCF-7 and T-47D cells [21]. This synergy was found to be associated with the differential effects of these two compounds on the expression of genes involved in cell survival and death, specifically downregulation of c-Myc and other anti-apoptotic proteins, including Bcl-xL and Mcl-1. In fact, cancers are heterogenous in cell type, with variation in metabolic pathways, and the effect of ribitol treatment is cell type-dependent; it is therefore desirable to investigate the combinational effect of ribitol with existing and potential drugs in different cancer cell types, to explore its therapeutic potential.

The objective of the current study is to identify more cancer cells that are sensitive to a defined combination treatment of ribitol and existing drugs by screening breast cancer cell lines. By analyzing the differential responses of cancer cells with different genotypes and phenotypes, together with alterations in their metabolism, critical metabolic pathways involved in ribitol-induced synergistic effects can be evaluated. Our results demonstrate the synergistic effects mediated by shikonin, a powerful regulator of cell death, in combination with ribitol against MCF-7 breast cancer cells. This effect is associated with inhibition of c-Myc, Mcl-1, and Bcl-xL by shikonin, and further enhanced by combined treatment with ribitol. In addition, our global metabolomics data analysis reveals that shikonin and ribitol alter metabolism differentially, which likely contributes to the enhanced disruption of energy supply and critical building elements. Altogether, the results emphasize the potential of selectively applying metabolites in combination with chemotherapeutic drugs to lower resistance and adverse toxicity, for more effective and safer cancer treatments.

## 2. Results

### 2.1. Drug Screening of Breast Cancer Cells Reveals Potential Targeted Therapeutic Candidates

We previously examined the potential synergistic effect of existing anti-cancer drugs with ribitol in breast cancer cells, and identified JQ1 as having significant synergism with ribitol in the triple-negative breast cancer cell line MDA-MB-231 [21]. Given the intrinsic heterogeneity of most cancers, as well as established cell lines in genotypes and phenotypes, the response of individual cell lines to drugs is likely highly variable. We therefore carried out screening of MCF-7 cells with various class of drugs (Table 1) targeting key enzymes of central carbon metabolism. Most of the drugs, except shikonin, did not reach the 50% reduction mark in terms of the viability of MCF-7 cells, even at the highest concentration tested, indicating drug resistance, as shown in Figure 1A,B. Chrysin, 2-DG, and honokiol, which are currently being used as supplements, reduced the viability of MCF-7 cells only at higher concentrations above 37 µM, 5 mM, and 25 µM, respectively. Among all the screened compounds currently being tested in clinical phases for the treatment of solid tumors, shikonin was the most effective in reducing the viability of MCF-7 cells in a dose-dependent manner, showing about a 50% viability reduction at 6.25 μM, and a reduction in viability of less than 30% and 10% at doses of 12.5 μM and 25 μM, after 72 h incubation, below the concentrations (EC_50_: 6.11 ±  0.12 μM) of the physiologically relevant dose in vivo [22] (Figure 2B).

We therefore selected shikonin for further tests in two other breast cancer cell lines: triple-negative breast cancer (MDA-MB-231) and HR-positive breast cancer (T-47D). The effect of shikonin at different concentrations on the three cell lines is shown in Figure 2. Shikonin treatment also reduced the viability of these two breast cancer cell lines, but with an EC_50_ of 13.41 ± 0.22 µM and 10.4 ± 0.12 µM for MDA-MB-231 and T-47D cells, respectively (Figure 2A). As illustrated in Figure 2B, this dose-dependent effect of shikonin on MCF-7 cells was clearly demonstrated with crystal violet staining and imaged by inverted microscopy.

### 2.2. Effect of Ribitol in Combination with Shikonin on Breast Cancer Cells

The synergy study of ribitol and shikonin was conducted on the three types of breast cancer cells (MCF-7, MDA-MB-231, and T-47D). As we reported earlier, ribitol alone, up to a concentration of 10 mM, did not exhibit a significant effect on cell viability in these cell lines [19]. The combination of ribitol and shikonin exhibited enhanced growth inhibitory effects over shikonin alone in the MCF-7 cancer cells as compared to other two breast cancer cells, again, in a cell-line and dose-dependent manner, as shown by surface matrix plots and crystal violet staining photomicrographs of MCF-7 cells (Figure 3 and Appendix A). The ATP CellTiter-Glo assay showed that the combination markedly enhanced shikonin’s effect of reducing the viability of MCF-7 cells across the dose range of ribitol, from 0.08 mM to 10 mM (Figure 3A). Synergy of the drug combination was also observed in the other two cell lines, but with a much narrower range of shikonin concentrations. In MDA-MB-231 cells, synergy was detected with shikonin concentrations ranging from 12.5 µM or less and a ribitol concentration above 0.63 mM (Figure 3B). Synergy was also detected in T-47D cells, but with an even more limited range of shikonin concentrations, above 6.25 µM, in combination with ribitol at a concentration between 0.31 mM and 2.5 mM (Figure 3C). Overall, the combination of ribitol and shikonin shows a strong significant synergistic effect when compared to shikonin alone on MCF-7 cells, but shows a much lower synergistic effect on the other two breast cancer cells. These results suggest that the amalgamation of the anti-cancer drug shikonin in combination of ribitol could be used for treating breast cancer selectively, but its cell type specificity requires further definition.

### 2.3. Combination of Ribitol and Shikonin Treatment Inhibits MCF-7 Cell Proliferation

To assess the impact of the treatments on cell growth, rather than static cell counts, we performed a growth curve analysis under live-cell imaging to determine the effects of single agents and their combinations (Figure 4). MuviCyte-Live cell imaging analysis revealed significantly higher growth inhibition of MCF-7 cells treated with the combination of shikonin (6 µM) and ribitol (5 mM), than with each agent alone (Figure 4A). The combined treatment showed a 40% and 25% reduction in cell proliferation compared to the control and shikonin alone, respectively. Based on these findings, we repeated the same experiment with a lower dose of shikonin (3 µM: half of the EC_50_ concentration) with ribitol (Figure 4B). This low concentration of shikonin with ribitol also showed significant (30%) growth inhibition of MCF-7 cells compared to shikonin alone. This test revealed that the synergistic effect occurred soon after the administration of the drugs, w and the effect was maintained until the conclusion of the experiment.

We then performed a wound-healing assay to determine the effect of ribitol (5 mM) alone and in combination with shikonin (3 μM) on the migration of MCF-7 cells. Once a scratch was made across the center of the wells, a live cell imaging system was used to record the movement of the cells migrating towards the gap over a period of 48 h (Figure 4C). The results showed that MCF-7 cells exhibited a >95% bridging of the acellular gap over the period, whereas the cells treated with ribitol alone and shikonin alone showed only partial bridging at this concentration. However, cells treated with the combination of ribitol and shikonin showed almost no bridging, a significant decrease in migratory activity when compared to the control, as well as to the single-agent treatment.

### 2.4. Ribitol Addition Augments Shikonin-Induced Apoptosis

To further explore the mechanisms involved in the drugs’ synergy, we investigated apoptotic pathways with annexin-V/PI co-staining experiments to determine the percentage of apoptotic cells (Figure 5). Only less than 5% cells were apoptotic in the untreated cells at 48 h. However, 11%, 26%, and 37% apoptotic cells were observed in the ribitol, shikonin, combination treatment groups, respectively (Figure 5A). A further significant increase in percentage (56%) of apoptotic cells was detected in the combined treatment group, compared 16% and 38% in the ribitol and shikonin single-treatment groups, respectively, after 72 h of exposure. These results suggest that ribitol increases the effects of shikonin-induced apoptosis.

To further investigate the involvement of various proteins known to play a crucial role in apoptosis, we measured the proteins c-Myc, p53, Mcl-1, and Bcl-xL, and compared the levels between the three treatment groups. Western blot analysis revealed that single-agent ribitol and shikonin treatment slightly upregulated the expression of the apoptosis master regulator p53 gene and downregulated the expression of the anti-apoptotic protein c-Myc. Expression of c-Myc and p53 was further downregulated and upregulated, respectively, in the cells treated with ribitol and shikonin in combination (Figure 5B). In addition, ribitol alone downregulated Mcl-1 expression, and, particularly, Bcl-xL expression. Shikonin produced a similar effect on Bcl-xL expression as that produced by the ribitol treatment but had a reduced effect on Mcl-1 expression (Figure 5C). These results support the conclusion that the combined ribitol and shikonin treatment arrests cell proliferation and induces apoptosis.

### 2.5. Effect of Ribitol and Shikonin on Metabolic Reprogramming of Breast Cancer Cells

Our prior study of the same MCF-7 cells with metabolomics analysis reported that ribitol treatment enhanced glycolysis and increased levels of GSH, but with a limited inhibitory effect on oxidative phosphorylation flux. This could explain the failure of ribitol to inhibit the growth of the cells, a potential consequence of greatly enhanced matriglycan expression in cancer cells [19]. Enhanced glycolysis is one of the main metabolic advantages of cancer cells which allows them to escape cell death, including apoptosis. To understand the potential contribution of the changes in metabolism induced by ribitol to the enhanced apoptosis observed in the combined treatment with shikonin, we also conducted global untargeted metabolomic profile analyses to determine the metabolic characteristics of the cells treated with shikonin in comparison with those treated with ribitol.

As reported previously, the level of glucose 6-phosphate (G6P), a metabolic intermediate shared by the pentose phosphate pathway (PPP) and glycolysis, was one of the most significant alterations after ribitol treatment [20] (Figure 6). The levels of G6P, however, were not significantly changed in the shikonin treated cells. The levels of 3-phopsphoglycerate (3-PG), further downstream molecules of the glycolysis pathway, decreased significantly (*p* < 0.05) in both treatment groups, with the shikonin-treated group showing greater reduction than the ribitol treatment group. Consistently with our earlier report, an increase in pyruvate and lactate levels was detected with ribitol treatment, whereas shikonin treated cells showed a significant decrease in pyruvate (*p* < 0.05) and lactate (*p* < 0.05) levels (Figure 6A). In contrast to the limited change with ribitol treatment, a significant decrease was detected in the levels of tricarboxylic acid cycle (TCA) cycle intermediates, such as citric acid (*p* < 0.05), α-ketoglutarate (α-KG) (*p* < 0.05), and succinic acid (*p* < 0.05) (Figure 6A), after shikonin treatment. These intermediates of the TCA cycle are used as building blocks for the synthesis of macromolecules, energy, and electron acceptors, which are important for cell survival and proliferation. Furthermore, ribitol treatment downregulated the levels of amino acids, particularly cysteine, proline, valine, and threonine, but not significantly, and isoleucine was downregulated significantly (*p* < 0.05) (Figure 6B) as compared to shikonin. Shikonin-treated cells also showed a significant decrease in one of the crucial intermediates of the glycolytic 3-PG pathway, which initiates major pathways like folate and nucleotide metabolism. Glycine and serine (*p* < 0.05) levels were also downregulated remarkably in comparison to the control (Figure 7A). Serine and glycine are biosynthetically crucial for cancer cells to undergo metabolic reprogramming to sustain growth and proliferation. Moreover, purine and pyrimidine metabolism, inosine monophosphate (IMP), and adenosine monophosphate (AMP) levels were also significantly (*p* < 0.05) downregulated after shikonin treatment, but remained unchanged after ribitol treatment (Figure 7A). Taken together, the dysregulation effects of ribitol on glycolysis and the TCA cycle could therefore enhance each other, promoting cell death by apoptosis in cancer cells, and these data support the notion that inhibition of the Warburg effect is involved in shikonin-induced cell death.

## 3. Discussion

Breast cancer is the most common cancer in women, and a leading cause of cancer-related death [23,24,25,26]. Breast cancer is a heterogeneous disease with diverse clinical behaviors, distinct biological properties, and distinct morphological patterns among various subtypes. In general, different cell types respond differently to different drugs based on their unique genotype and phenotype profiles, and different cell populations within a tumor mass can exhibit varying responses to the same drug, due to differences in their microenvironment and gene expression profiles. Thus, distinct molecular signatures are crucial in mediating cell response to treatments and in defining drug targets. Earlier studies suggest that this principle is also applicable to antimetabolic drugs, especially in combination treatment. Heterogeneity in metabolism has been reported in a broad range of cancers, and specifically in different breast cancers, by multi-omics analysis, indicating a likelihood of differential responses when antimetabolic drugs and metabolite intervention are considered as treatments [6,27,28,29,30,31,32]. Current treatments are associated with severe side effects, and cancer cells often develop resistance. Efforts to overcome these barriers by combined drug treatment have been practiced with notable success for some cancers, including breast cancer. However, most combination treatments use drugs that directly target the cell cycle and proliferation, with high toxicity when used individually. Exploring a new class of compounds for combinatorial treatment with low toxicity is therefore highly desirable [33,34,35,36]. Recent advances in techniques of metabolomics have greatly improved our understanding of metabolic characteristics unique to individual cancers. Similarly, the effect of existing and potential cancer drugs on cell metabolism can now be investigated with relative ease, and the development of novel combinational therapies targeting crucial metabolic pathways offers an alternative strategy against breast cancer.

Among the drugs we tested in this study, shikonin showed the highest effect at the lowest concentrations on breast cancer cells, compared to the other drugs. Shikonin, a naphthoquinone compound originally extracted from the root of Lithospermum erythrorhizon, is extensively reported to exert anti-tumor activity against various types of cancer [37,38,39,40,41]. The anti-cancer effect of shikonin is mainly achieved by the inhibition of malignant cell growth and the induction of caspase-3-dependent apoptosis, DNA cleavage, and cell cycle arrest, likely through its accumulation in mitochondria and alteration of cellular Ca2+ and reactive oxygen species (ROS) [39,42,43,44]. Shikonin also inhibits cell proliferation through the alteration of glucose and lactate metabolism, by targeting tumor pyruvate kinase M2 (PKM2) [39,43,44,45,46,47]. Preclinical studies have shown that this class of drugs targeting PKM2 is effective against some cancer types, and early clinical trials have shown promising results, but with dose-dependent toxicity in patients [46,48,49,50].

Shikonin has also been tested for treating breast cancer. It was reported earlier that in luminal breast cancer cells, shikonin inhibits estrogen-stimulated cell proliferation without affecting normal human mammary epithelial cells [51]. Previous reports have stated that shikonin inhibits the cell growth and proliferation of SK-BR-3 cells [52]. In addition, when combined with other pharmaceuticals, such as paclitaxel, metformin, and tamoxifen, shikonin exhibits a synergistic effect and enhances the sensitivity of cancer cells to the drugs [53,54,55]. Shikonin improves tamoxifen’s antiproliferative efficacy in hormone-sensitive breast cancer cells. Differently from the combination treatment described in these studies, we examined the use of a metabolite ribitol for enhancing the anti-cancer effect of shikonin. Our results support earlier findings that shikonin is highly effective in inhibiting the growth and proliferation of breast cancer cells, especially the luminal type of MCF-7 cells [19]. This effect is greatly enhanced in the presence of ribitol. Perhaps more importantly, the synergistic effect is achieved at significantly lower doses of shikonin. Since ribitol is a natural metabolite, and has been safely used with doses up to 15 g a day for more than 1 year in clinical trial [56], this would minimize the barrier imposed by the dose-related toxicity of shikonin for cancer treatment in clinical settings.

The mechanisms by which ribitol enhances the growth inhibition and death of the cancer cells are not entirely clear. Apoptosis is programmed cell death that plays a crucial role in both physiological and pathological conditions [57,58,59,60]. Emerging evidence has suggested that the aberrant expression of survival factors protects tumor cells from death, following the activation of intrinsic and extrinsic apoptotic pathways, especially in oncogenesis [59,61]. c-Myc, p53, and Bcl-2 family proteins play a critical role in determining cell death and survival. Previous studies have demonstrated that shikonin exerts its anti-tumor effects by modifying programmed cell death, including apoptosis [62,63,64,65,66]. In this study, we found that shikonin, along with ribitol, induced apoptosis in breast cancer cells. This is associated with downregulation of c-Myc and upregulation of p53. Activation of the c-Myc proto-oncogene activates genetic programs that promote cancer growth and proliferation. This is consistent with our results showing a reduction in the c-Myc expression and inhibition of cell growth and migration [67,68]. Upregulation of p53 by shikonin would be consistent with the observed increase in cell death by apoptosis. Furthermore, consistently with the role that p53 plays in cell death and survival, the two critical anti-apoptosis members of the Bcl-2 family proteins, Bcl-xL and Mcl-1, were downregulated. Ribitol enhances this effect exerted by shikonin, which is consistent with the enhanced apoptosis and cell death observed with their combined treatment on cells.

Enhanced glycolysis and glutaminolysis and elevated fatty acid and nucleotide synthesis are generally considered to be the hallmarks of MYC-driven cancer growth and metastasis. This is in agreement with our metabolomics analysis. Shikonin treatment significantly decreased the levels of pyruvate and lactate, as well as the TCA cycle intermediates citric acid, α-ketoglutarate (α-KG), and succinic acid. A greater reduction was also detected in the levels of 3-phopsphoglycerate (3PG). This will greatly diminish the cells’ sources of energy and constituents, critical factors for the maintenance of cell survival and growth. In contrast, the main effect of ribitol treatment alone is the enhancement of glycolysis, as indicated by the increase in G6P and F6P levels, although the TCA cycle is also disturbed. This is consistent with its innocuous effect on the cells. We hypothesize that the enhancement of glycolysis by ribitol provides sufficient energy for the cells to overcome any detrimental effect caused by limited disruption to the TCA cycle. However, shikonin significantly blocks the generation of 3PG, which is the critical intermediate for ATP production by glycolysis. Thus, combination of the two leads to a more severe reduction in the supply of energy and essential constituents, including AMP and IMP, from both pathways, consequently leading to enhanced growth arrest and cell death. It remains to be investigated how this metabolic effect of ribitol is related to the enhanced reduction in the expression of c-Myc, Mcl-1, and Bcl-xL induced by shikonin.

In summary, our results show that ribitol synergizes with shikonin to produce increased cell death and decreased cell migration. Both drugs exert a similar effect on the expression of c-Myc and p53, leading to enhanced apoptosis (Figure 7B). This process is associated with alterations in glycolysis (mainly by ribitol) and inhibition of the TCA cycle (by shikonin). Modulation of metabolic pathways by metabolites represents a novel approach to sensitize cell response and overcome tumor resistance. The primary advantage of the use of ribitol and shikonin in combination is the potential for significant reduction in therapeutic dosages, thereby mitigating adverse effects for clinic applications. It should be noted that while ribitol is a metabolite found in nature and has been shown to be safe in animal models and phase III clinical trials, the synergistic anti-cancer effect of ribitol is highly selective in partner drugs and in target cells. Furthermore, such an effect clearly requires validation by in vivo animal model experiments. A much-detailed mechanistic exploitation is now warranted.

## 4. Materials and Methods

### 4.1. Cell Lines and Culture

The human breast cancer cell lines MCF-7 (ATCC-HTB-22), T-47D (ATCC-CRL2865), and MDA-MB-231 (ATCC-HTB-26) were purchased from ATCC (Manassas, VA, USA). MCF-7 and T-47D were cultured in DMEM-GlutaMAX (Life technologies, Carlsbad, CA, USA) plus 10% fetal bovine serum (FBS 10082-147, R&D systems) and 10 µg/mL insulin (I5500 Sigma Aldrich, St. Louis, MO, USA). MDA-MB-231 was grown in DMEM-GlutaMAX + 10% FBS at 370C in a 10% CO_2_ incubator. The culture medium was obtained from Gibco by Life Technologies.

### 4.2. Compound Screening

In primary screening, cells were pretreated with or without 5 mM ribitol for 72 h in plates. Then, 10,000 cells per well were seeded in a 96-well plate. We screened a panel of drugs, such as shikonin (Selleckchem, Houston, TX, USA), Gemcitabine (EMD Millipore, Billerica, MA), GSK2837808A, Chrysin, Dichloroacetate, BPTES, Honokiol, CHS828, FK866, and chrysin, purchased from Sigma Aldrich (St. Louis, MO, USA). Cancer cell lines were treated with a single agent with various concentrations. Cell viability was determined after 72 h, following the manufacturer’s instructions (CellTiter-Glo reagents, Promega, Madison, WI, USA). Viability was measured with a Biotek Synergy NEO Multi-Label Reader (Agilent, Santa Clara, CA, USA), as a percentage of the response relative to both cells treated with DMSO alone (0% response). Compounds with both low response (≤mean + 3 SEM cutoff) in cells not treated with ribitol, and high response (≥mean + 3 SEM cutoff) in cells treated with ribitol, were considered hits. The Z-factor for all the plates was greater than 0.9, demonstrating the robustness of the assay.

### 4.3. Cell Viability Assay and Combenefit Synergy Software 2.021 

In our screening, as shikonin was identified as one of the top drugs involved in synergistic combinations with ribitol, it was chosen as the anchor drug. Cell lines were treated for 72 h with ribitol in single application and in combination with shikonin in all possible combinations of doses. Cell viability was determined after 72 h using CellTiter-Glo (Promega, Madison, WI, USA), following the manufacturer’s instructions. These results were analyzed using Combenefit software (2.021). We configured a 96-well microplate assay compatible with the Combenefit dual-drug interaction software; cell concentration data, read spectrophotometrically, were submitted to rigorous statistical analysis for synergistic or antagonistic interactions, calculated according to the Loewe additivity model, which is part of the Combenefit package. Combenefit software was used to perform synergism; it calculates and displays the synergism–antagonism distributions and computes a variety of metrics from the distributions. The dose–response curve for each of the chemotherapeutic drugs was computed by the software from all biologic replicates of that drug combination.

### 4.4. Proliferation Assay

The antiproliferative activity of ribitol and shikonin was assessed by the MuviCyte™ Live-Cell Imaging System. Breast cancer cells seeded in 24-well culture plates in complete medium were incubated overnight. After overnight incubation, cells were then grouped into control (untreated), ribitol, shikonin, and the combination of ribitol and shikonin. Cells were incubated for 48 h in a MuviCyte (PerkinElmer, Waltham, MA, USA) Live-Cell imaging system attached to an incubator, at 37 °C in 5% CO_2_. The cell migration was monitored by the MuviCyte™ Live-Cell Imaging System (PerkinElmer, Waltham, MA, USA) for 48 h. Cell growth plots were captured at 2 h intervals via the MuviCyte Live-Cell imaging software version 2.0.26. A migration assay was also performed after MCF-7 cells were treated with ribitol and shikonin for 72 h, by a wound healing assay using the MuviCyte™ Live-Cell Imaging System.

### 4.5. Real Time-Glo™ Annexin V Apoptosis Assay

The purpose of this experiment was to determine the apoptotic activities of combination therapy in comparison to monotherapy. Briefly, MCF-7 cells were treated for 48 h and 72 h with half of the EC_50_ of shikonin and 5mM ribitol. In the Real Time-Glo™ Annexin V Apoptosis assay, time- and dose-dependent increases in luminescence were produced by the annexin V fusion protein binding, which preceded temporal increases in fluorescence via loss of membrane integrity. In this experiment, we analyzed the response of the breast cancer cells treated with ribitol (5 mM) and shikonin (3 µM) over 48 h. Readings were measured on a BioTek Synergy NEO Reader (Agilent, Santa Clara, CA, USA) as a percentage of the response relative to cells treated with DMSO alone (0% response).

### 4.6. Western Blot

The level of apoptotic proteins was assessed by subjecting 60 µg of each total cell lysate to immunoblot analysis. Cells were lysed in Triton lysis buffer containing 1% Triton X-100, 50 mM Tris pH 8, 150 mM NaCl, 1 mM EDTA, and 1× Protease Inhibitor Cocktail (Sigma). After clarification of the lysates by centrifugation at 13,000 rpm for 10 min at 40 °C, the protein concentration of the lysates was measured using the Bradford method (BioRad, Hercules, CA, USA). Samples were then electrophoretically separated on a 4–15% Criterion Tris-HCI 18-well gel, (3450028, Bio-Rad Laboratories, Hercules, CA, USA) and transferred onto a supported nitrocellulose membrane. Immunoblots were probed with primary antibodies c-Myc (SC-40 Santa Cruz Biotechnology, Santa Cruz, CA, USA), p53 (SC-126 Santa Cruz), Bcl-xL (AB32370 Abcam, Cambridge, UK), and Mcl-1 (SC-12756 Santa Cruz Biotechnology, Santa Cruz, CA, USA) at 1:1000 dilution in 5% nonfat dry milk/1xTBS-0.05% Tween. Rabbit polyclonal antibody to actin (A2066 Sigma) was used at 1:3000 dilution in 5% nonfat dry milk/1xTBS-0.05% Tween as a loading control. The blots were incubated with primary antibody overnight at 4 °C. After washing, membranes were subsequently incubated with secondary antibodies of HRP-conjugated goat anti-mouse IgG (1:3000), or goat anti-rabbit IgG (1:3000), in their blocking buffer for 1 h 30 min. Bands were detected using the ECL detection Kit NEL 104001EA (PerkinElmer) on blue basic autoradiography film (USA Scientific, Ocala, FL, USA).

### 4.7. Metabolomics

Cell pellets from the 72 h cultures treated with ribitol, shikonin, or the untreated control were collected via scraping and centrifugation, after PBS washing, and submitted to the West coast metabolomics center (WCMC) core for metabolomic analysis. Primary metabolomic analysis was performed with GCTOF-C in a resuspension volume of 100 µL, with an injection volume of 0.5 µL. A total of 126 metabolites were identified after data processing, with the study data normalized to the average mTIC for the respective sample type. Key metabolites (37) were identified from these identified metabolites, and the fold change (Log2) and statistical significance were calculated between experimental groups by *t*-tests.

### 4.8. Statistical Analysis

All data are expressed as the mean ± SEM of three independent experiments performed in triplicate. The differences between two groups were evaluated by using Student’s *t*-test for unpaired observations. Individual means were compared using unpaired *t*-tests. A *p* < 0.05 was considered statistically significant.

## Figures and Tables

**Figure 1 ijms-26-02661-f001:**
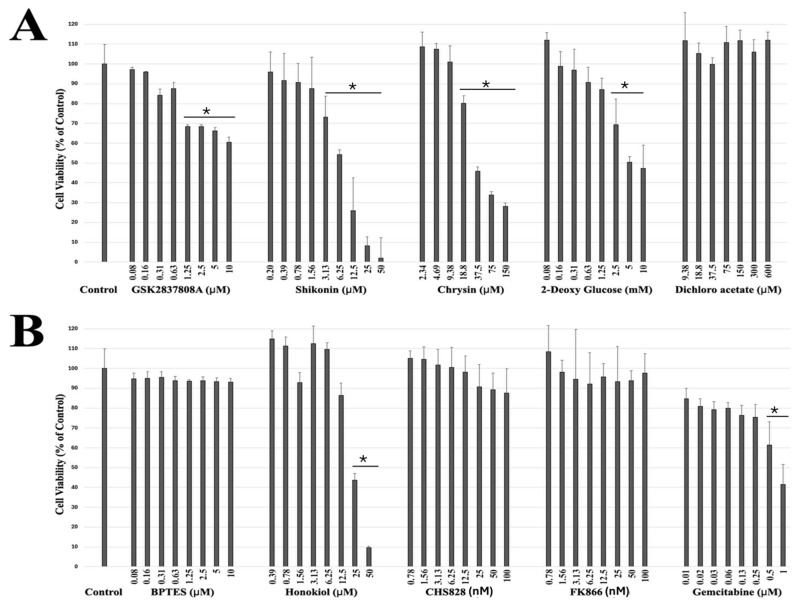
Primary screening of anti-cancer compounds on breast cancer cells. Representative dose–response curves of MCF-7 cells treated with (**A**) (i) GSK2837808A, (ii) Shikonin, (iii) Chrysin, (iv) 2-deoxy D-glucose, (v) Dichloroacetate, (**B**) (i) BPTES, (ii) Honokiol, (iii) CHS828, (iv) FK866, and (v) Gemcitabine, for 72 h; viability was measured using ATP CellTiter-Glo assay. Each value represents average of independent experiments with triplicate determinations. Data were calculated from triplicate experiments and error bars represented as mean ± SEM. * *p* < 0.05 compared to untreated control.

**Figure 2 ijms-26-02661-f002:**
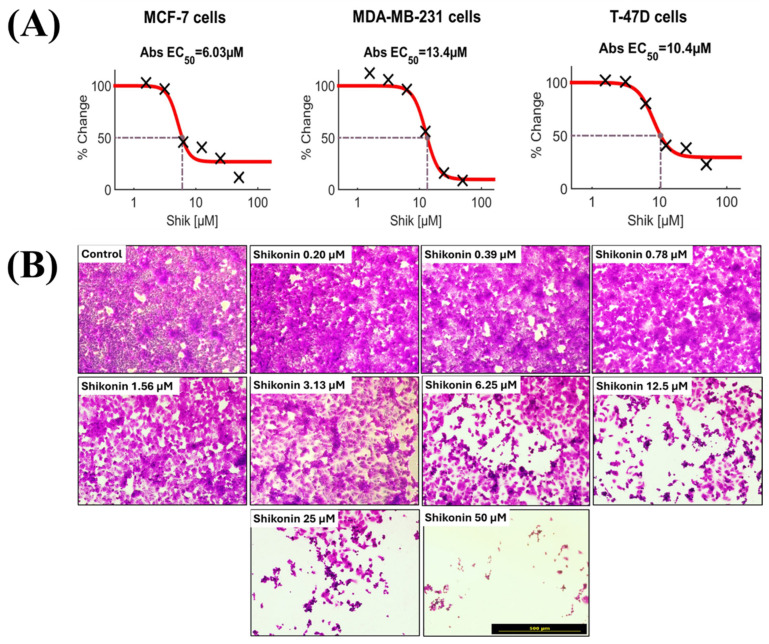
Effect of shikonin on breast cancer cells. (**A**) Dose–response curves of breast cancer cells MCF-7, MDA-MB-231, and T-47D treated with shikonin for 72 h, with viability measured using ATP CellTiter-Glo assay. (**B**) Representative dose–response images of MCF-7 cells treated with shikonin for 72 h and photographed after crystal violet staining. Images were taken by inverted light microscopy at ×100 magnification (scale bar = 500 µm).

**Figure 3 ijms-26-02661-f003:**
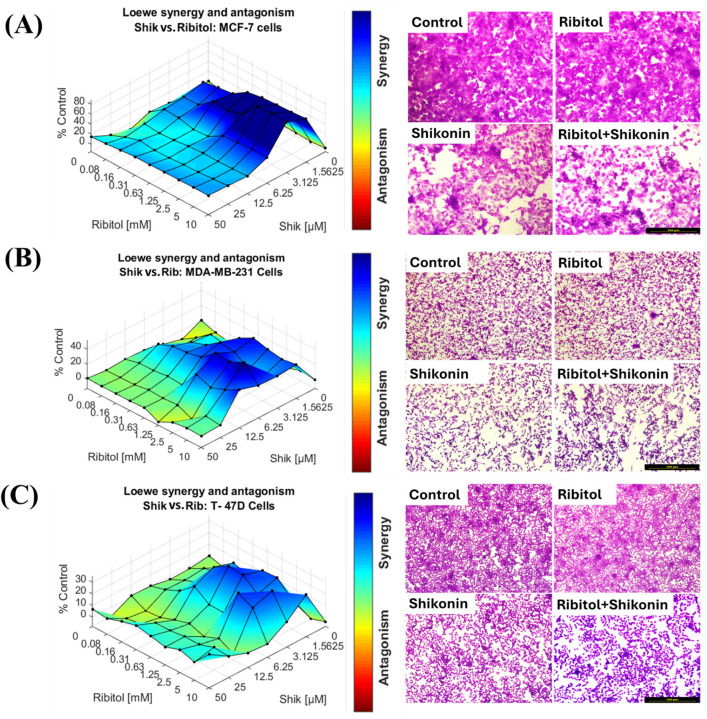
Synergistic effects of ribitol and shikonin on three types of breast cancer cells. Synergistic effect of ribitol and shikonin on (**A**) MCF-7, (**B**) MDA-MB-231, and (**C**) T-47D cells after 72 h incubation; viability was measured using ATP detection (CellTiter-Glo assay). Blue color indicates synergy between drugs. Synergy between ribitol and shikonin was analyzed using Combenefit software (version 2.021). Along with surface matrix plots, breast cancer cells were stained with crystal violet after shikonin treatment for 72 h, and images were taken by inverted light microscopy at ×100 magnification (scale bar = 500 µm).

**Figure 4 ijms-26-02661-f004:**
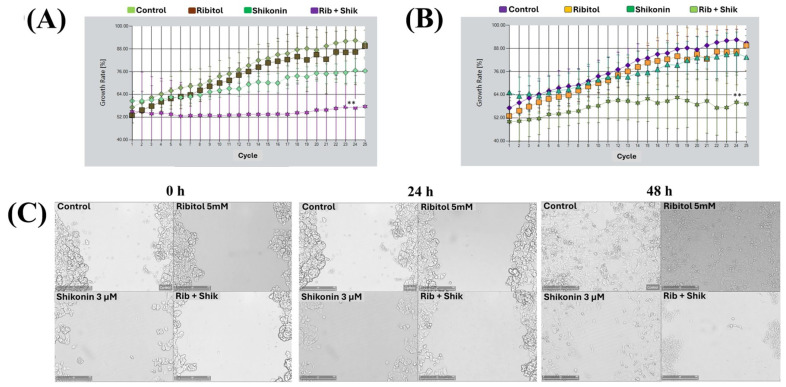
The combination of ribitol and shikonin inhibits the proliferation of MCF-7 breast cancer cells. (**A**) The treatment of cells with ribitol (5 mM) and shikonin (6 µM; EC_50_ concentration) over a period of 48 h and (**B**) The treatment of cells with ribitol (5 mM) and shikonin (3 µM; half of the EC_50_ concentration) over a period of 48 h. Untreated cells were used as a control. (**C**) Representative images of the migration assay after the cells were treated with ribitol and shikonin for 48 h. Live cell images at 0 h, 24 h, and 48 h are illustrated at ×100 magnification (scale bar = 250 µm). The error bars represent the mean ± SEM. *p* < 0.05 is considered significant. ** *p* < 0.01 compared with the untreated control.

**Figure 5 ijms-26-02661-f005:**
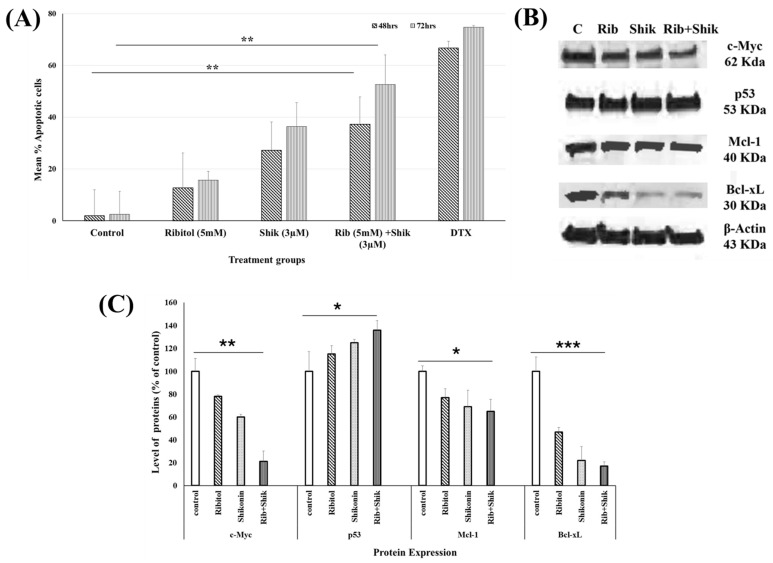
Potential anti-tumor activity of ribitol and shikonin combination on induction of apoptosis in breast cancer cells. (**A**) Representative bar graph shows percentages of apoptotic cells as measured by Annexin-V staining of MCF-7 cells 48 h and 72 h after exposure to ribitol and shikonin. Docetaxel (DTX) was used as positive control. Bars represent average of three biological replicates. (**B**) Western blot of cellular apoptotic proteins of MCF-7 cells treated with ribitol and shikonin. Representative images of Western blot analysis of various pro/anti-apoptotic proteins: p53, c-Myc, Bcl-xL, and Mcl-1. Values were normalized to housekeeping gene β-actin expression. (**C**) Quantification of apoptosis-related proteins in breast cancer cells after treatment. Intensity of signals was quantified by densitometric analysis. Data were calculated from triplicate experiments. Error bars represent mean ± SEM. *p* < 0.05 is considered significant. * *p* < 0.05, ** *p* < 0.01, *** *p* < 0.001 compared with untreated control.

**Figure 6 ijms-26-02661-f006:**
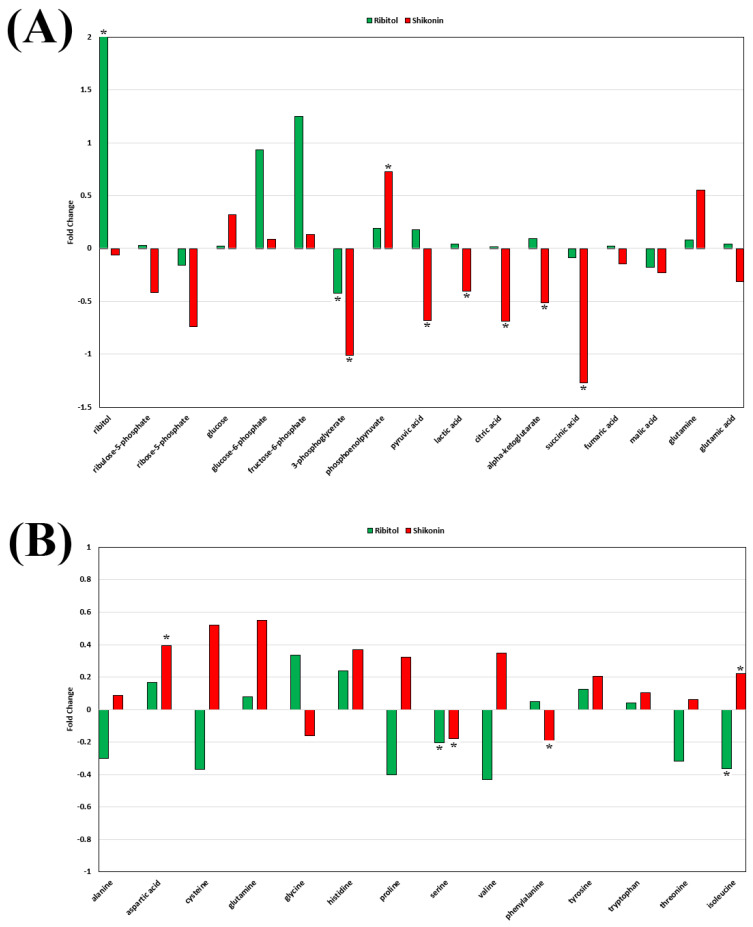
Ribitol and shikonin perturb key metabolites of glucose metabolism in MCF-7 cells. (**A**) Alterations in key metabolites of glycolysis and TCA cycle, such as 3-phophoglycerate, pyruvate, lactate, citrate, α-Ketoglutarate, and succinate levels, were downregulated significantly (*p* < 0.05) after shikonin treatment. (**B**) Alterations in levels of amino acids like glycine and serine were downregulated after shikonin treatment, and cysteine, proline, valine, threonine, and isoleucine (*p* < 0.05) levels were downregulated after ribitol treatment. Data were calculated from triplicate experiments. * *p* < 0.05 is considered significant.

**Figure 7 ijms-26-02661-f007:**
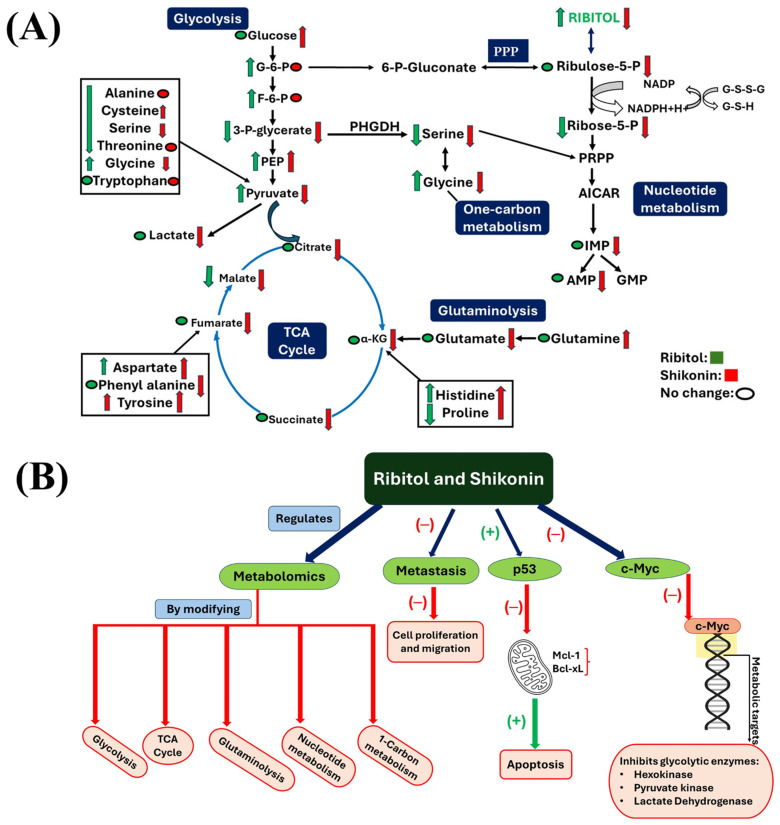
Metabolomics of MCF-7 cells treated with ribitol and shikonin. Key metabolite alterations of each pathway of central carbon metabolism, shown in schematic way. (**A**) Shikonin treatment downregulated key intermediates of glycolysis (3-PG, pyruvate and lactate), TCA cycle (Citrate, α-KG, succinate, fumarate and malate), PPP (Ribose-5-p), and nucleotide metabolite (IMP and AMP) levels. Shikonin also downregulated levels of glutamate, as well as glycine and serine. Ribitol particularly downregulated 3-phosphoglycerate and serine. (**B**) Overall view of key metabolite alterations of MCF-7 cells treated with ribitol and shikonin within established pathways. Red-colored arrows represent changes with shikonin treatment, and green-colored arrows represent changes with ribitol treatment. Downregulated metabolites are indicated by down-pointing arrows, and upregulated metabolites by up-pointing arrows. Ribitol and shikonin combination regulates metabolomics, inhibits proliferation and migration of cells, and induces apoptosis, by targeting key enzymes of glycolysis. G-6-P: glucose-6-phosphate; F-6-P: fructose-6-phosphate: PEP: phosphoenolpyruvate; 3-PG: 3-phosphoglycerate; α-KG: α-ketoglutarate; PPP: pentose phosphate pathway; PRPP: phosphoribosyl pyrophosphate; AICAR: 5-aminoimidazole-4-carboxamide ribonucleotide; IMP: inosine 5′-monophosphate; AMP: adenosine monophosphate; GMP: guanosine monophosphate. (−): inhibition; (+): stimulation.

**Table 1 ijms-26-02661-t001:** Compounds used to examine their anti-cancer efficacy against breast cancer cells.

S. No	Drug Name	Target
1	GSK2837808A	Lactate Dehydrogenase Inhibitor II
2	Shikonin	Pyruvate kinase M2 (PKM2) inhibitor
3	Chrysin	Targets succinate dehydrogenase (SDH)
4	2-Deoxy-D-glucose	Glycolytic inhibitor with antiviral activity.
5	Dichloroacetate	Mitochondrial pyruvate dehydrogenase kinase (PDK) inhibitor
6	BPTES	Glutaminase GLS1 (KGA) inhibitor
7	Honokiol	Inhibitor of complexes I, II, and V of ETC
8	CHS828	Inhibitor of nicotinamide phosphoribosyltransferase (NAMPT)
9	FK866	Nicotinamide phosphoribosyltransferase inhibitor
10	Gemcitabine, HCl	Ribonucleotide Reductase Inhibitor II

## Data Availability

The datasets generated and/or analyzed during the current study are available from the corresponding author upon reasonable request.

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
