# Peer review of "Synergistic Effect of Ribitol and Shikonin Promotes Apoptosis in Breast Cancer Cells"

_ijms, 2025, doi:10.3390/ijms26062661_

Round 1

Reviewer 1 Report

Comments and Suggestions for Authors

Reviewer #1

Journal: International Journal of Molecular Sciences (ISSN 1422-0067)
Manuscript ID: ijms-3403596
Type: Article
Title: Synergistic Effect of Ribitol and Shikonin Promotes Apoptosis in Breast Cancer Cells
Authors: Ravi Doddapaneni*, Jason D. Tucker, Pei J. Lu, Qi L. Lu*

Doddapaneni et al. in this study claim that combination treatment with Ribitol and Shikonin has a synergistic effect on apoptosis in breast cancer cell lines, which is due to the negative effects on the glycolytic process and the TCA cycle. This effect is purportedly mediated through the inhibition of glycolytic activity and disruption of the tricarboxylic acid (TCA) cycle. Ribitol, a relatively obscure metabolite, is purportedly linked to laminin-binding glycan (matriglycan), which has been associated with high-grade malignancies and poor prognosis in breast cancer. Consequently, the investigation of this metabolic pathway and its therapeutic implications in the context of breast cancer is of notable scientific and clinical interest.

Notwithstanding the study’s promising premise, numerous issues present in the manuscript raise significant concerns regarding its readiness for publication in IJMS. Addressing the following issues is imperative for advancing this manuscript towards a potential publication.

Main Review

1. Figure 1A and B:
Considering the rigor and standards expected at the journal level, the author should refrain from relying solely on bar graphs to substantiate the data presented. Given that the author indicates all drugs were administered for a 72-hour treatment period, it would greatly enhance the credibility of the findings to include representative images of cell staining (e.g., using Giemsa, Crystal Violet, or Kumasi) in the figure. This would provide visual corroboration of the experimental results and strengthen the overall reliability of the data. Furthermore, as the experiment should have been conducted with a minimum of three independent replicates, the raw data should be provided as an Excel file for transparency and reproducibility. Additionally, the inclusion of appropriate statistical significance markers on the graphs is essential for clearly conveying the robustness of the results.

2. Figure 2:
While the author may believe that presenting both the left and right panels of the figure provides additional clarity, it ultimately introduces ambiguity, as both panels depict essentially the same data. I strongly recommend removing one of these panels to streamline the figure and enhance its interpretability. Instead, the author should consider including a representative photomicrograph of cell lines from the experiment to provide visual context for the data. Additionally, there is insufficient explanation regarding the numerical value corresponding to the color scale in the upper-right corner of the left panel. This should be clearly defined to ensure that readers can accurately interpret the data. A more thorough legend or annotation is warranted to elucidate the meaning of this indicator color.

3. Figure 4:
Figure 4B indicates that the combination of Ribitol and Shikonin results in a decrease in the protein levels of c-Myc, Mcl-1, and Bcl-Xl, while simultaneously increasing the expression of p53. However, the interrelationship between these proteins is not immediately clear, given the complex signaling networks that mediate their interactions. The authors should provide a more detailed explanation of the mechanisms by which Ribitol and Shikonin exert their effects, particularly in relation to key apoptotic signaling molecules such as Bcl-2, Bax, Bak, t-Bid, caspase-9, caspase-3, and PARP-1, which are central to mitochondrial-mediated cell death pathways.

In particular, c-Myc and p53 are well-established oncoproteins and tumor suppressor proteins, respectively, with mechanisms of action that are distinct from those of the apoptosis-related proteins. The authors should carefully elaborate on how these proteins interact with additional modulators, such as p21, NOXA, PUMA, and MDM2, to clarify their role in the observed effects.

Moreover, to substantiate the proposed mechanism of action, it would be valuable to assess whether cell death induced by the Ribitol and Shikonin combination is attenuated upon temporary inhibition of key genes such as c-Myc, p53, Mcl-1, or Bcl-Xl, using RNA interference (RNAi) or gene knockdown/knock-out approaches. This would provide further insights into the causal relationship between these molecular targets and the observed phenotypic outcomes.

Lastly, since Figures 4B and 4C present the same data, I recommend consolidating these figures and incorporating the aforementioned mechanistic details into Figure 4C for clarity and conciseness.

4. Figure 5:
As previously highlighted, the authors emphasize the synergistic anticancer effects of the Ribitol and Shikonin combination. However, to substantiate this hypothesis, the current presentation of Figure 5 has several notable shortcomings. Specifically, the analysis of cellular metabolic proteins related to the glycosis or TCA cycle in the context of Ribitol and Shikonin treatment—when compared to treatments with Ribitol alone or Shikonin alone—introduces several inconsistencies and contradictions in the data interpretation.

To properly validate the proposed mechanism of synergy, the experimental design must be expanded to include a combination treatment group. In particular, if the authors wish to claim that cell death induced by the Ribitol and Shikonin combination is linked to mitochondrial-related metabolic alterations, this assertion must be rigorously demonstrated by including controls and all relevant treatment groups (control, Ribitol, Shikonin, and the combination of Ribitol and Shikonin). A more comprehensive analysis will provide clarity and strengthen the credibility of the hypothesis.

The authors should exercise particular caution when introducing novel anticancer candidates and discussing their effects in a manner that may appear superficial or inadequately substantiated. Such practices not only undermine the quality of the research but also pose potential risks to the integrity of global collaborative research endeavors. Rigorous experimental validation and a more detailed, transparent presentation of the data are essential for maintaining scientific credibility.

5. Figure 6:
The rationale underlying the schematic model presented in Figure 6 remains unclear. The author does not provide sufficient justification for how the relationships between key proteins, such as c-Myc, p53, Mcl-1, and Bcl-Xl—established through experiments in Figure 4—are incorporated into this model. As it stands, this figure appears to be a speculative abstract hypothesis rather than a data-driven representation of the experimental findings.

There is a lack of consistency between the experimental content presented in the manuscript and the schematic model proposed in Figure 6. If the author intends to emphasize that Ribitol and Shikonin exhibit distinct mechanisms of action, which may contribute to their observed synergistic effects, this should be more clearly articulated. Furthermore, the author should consider the clinical implications of these candidate anticancer agents. Has the potential for side effects been adequately evaluated, particularly in the context of their application to breast cancer patients? It is crucial for the author to maintain a patient-centered perspective, particularly when proposing new therapeutic candidates.

In summary, the schematic model in Figure 6 should be directly informed by the experimental data presented in the manuscript. It is essential that the model be grounded in the findings thus far, and the author must clarify the rationale behind its construction. I urge the authors to revise this figure to better reflect the empirical evidence and to provide a more coherent and scientifically justified explanation.

Minor Review

Question 1: Introduction Section

In the 'Introduction' section, the authors describe that the combination of Ribitol and JQ1 demonstrated a prominent synergistic effect in the MDA-MB-231 breast cancer cell line, but no such effect was observed in the hormone-positive breast cancer cell lines, MCF-7 and T47D. If the current study claims that the combination of Ribitol and Shikonin had a similar effect across all cell lines, the authors must provide an explanation for this apparent discrepancy. Specifically, if the observed differences are due to a hormone-dependent event, this should be clearly discussed in the 'Discussion' section. The authors should incorporate relevant literature to support this explanation, as the introduction has already alluded to such a possibility. A more comprehensive clarification of the underlying mechanisms contributing to the observed differential effects would strengthen the manuscript.

Question 2: Reference Citation

On page 9, line 203, there is still the word 'ref' that seems unnecessary.  Such mistakes cast doubt on the author's attention to detail and commitment to scholarly rigor. This should be rectified promptly. Furthermore, when making claims, particularly in the 'Introduction' and 'Discussion' sections, it is essential to include at least 2-5 references to substantiate the content. References are critical for bolstering the scientific credibility and impact of the work. Without proper citation, the manuscript may lose its trustworthiness. The authors should ensure that all necessary references are included to appropriately contextualize their findings and support the claims made throughout the manuscript.

Comments on the Quality of English Language

N/A

Author Response

We would like to thank the reviewers for their comments and suggestions to help us strengthen the manuscript. We appreciate the encouraging comments made on the significance of our findings towards a more effective treatment for this lethal tumor. Please see below our response to each of the reviewers’ points and the modified text accordingly.

Reviewer #1

Main Review

  1. Comment:

Figure 1A and B:
Considering the rigor and standards expected at the journal level, the author should refrain from relying solely on bar graphs to substantiate the data presented. Given that the author indicates all drugs were administered for a 72-hour treatment period, it would greatly enhance the credibility of the findings to include representative images of cell staining (e.g., using Giemsa, Crystal Violet, or Kumasi) in the figure. This would provide visual corroboration of the experimental results and strengthen the overall reliability of the data. Furthermore, as the experiment should have been conducted with a minimum of three independent replicates, the raw data should be provided as an Excel file for transparency and reproducibility. Additionally, the inclusion of appropriate statistical significance markers on the graphs is essential for clearly conveying the robustness of the results.

Response: We appreciate the Reviewer’s point. According to the Reviewer’s suggestion, we have added crystal violet staining microscopic images of MCF-7 cells treated with ribitol and shikonin.  Now, we divided figure 1 as figure 1 (A&B) and figure 2 (A&B) due to inclusion of large size of microscopic cell staining images. As per reviewer suggestion, we attached raw data of excel sheet file of showing three independent replicates of each experiment readings by luminometer. Also, we updated statistical significance on the graphs in Figure 1 and modified accordingly in figure legend (highlighted in text).

  1. Comment:

Figure 2:
While the author may believe that presenting both the left and right panels of the figure provides additional clarity, it ultimately introduces ambiguity, as both panels depict essentially the same data. I strongly recommend removing one of these panels to streamline the figure and enhance its interpretability. Instead, the author should consider including a representative photomicrograph of cell lines from the experiment to provide visual context for the data. Additionally, there is insufficient explanation regarding the numerical value corresponding to the color scale in the upper-right corner of the left panel. This should be clearly defined to ensure that readers can accurately interpret the data. A more thorough legend or annotation is warranted to elucidate the meaning of this indicator color.

Response: We would like to thank the reviewer for the suggestion. As per reviewer advice, we now removed the dose response synergy matrix in figure 2 (moved to supplementary information as supplementary figure S1 with legend).  Along with surface matrix plots, for better interpretation now included crystal violet staining photomicrographs of various types of breast cancer cells treated with ribitol and shikonin. The figure (Figure 3) and its legend was modified accordingly and incorporated in text as well (highlighted in text).

Dose response synergy matrix figures were moved to supplementary information and numerical values were explained corresponding to the color scale.  Combenefit analysis revealed the synergy between drugs and showed a synergistic effect of the combination therapy in MCF-7 cells at the doses ranges from 1.56 µm - 6.25 µm shikonin with 0.6mM-10mM ribitol (Blue color indicates synergy). The same analysis was also conducted for MDA-MB-231 and T-47D cells but showed a low degree of synergism or no synergistic effect at most of the doses (Green color).

Regarding numerical value in synergy matrix, the synergy score, as the average excess response to drug interactions in cells, >20 indicates synergistic effect. For MCF-7 cells, the score was more than 40 (doses ranges from 1.56 µm-6.25 µm shikonin with 0.6mM-10mM ribitol) which indicates strong synergy between drugs. Negative value means no synergy between drugs. These explanations are now added to the supplementary figure (S1).

  1. Comment:

Figure 4: Figure 4B indicates that the combination of Ribitol and Shikonin results in a decrease in the protein levels of c-Myc, Mcl-1, and Bcl-Xl, while simultaneously increasing the expression of p53. However, the interrelationship between these proteins is not immediately clear, given the complex signaling networks that mediate their interactions. The authors should provide a more detailed explanation of the mechanisms by which Ribitol and Shikonin exert their effects, particularly in relation to key apoptotic signaling molecules such as Bcl-2, Bax, Bak, t-Bid, caspase-9, caspase-3, and PARP-1, which are central to mitochondrial-mediated cell death pathways.

In particular, c-Myc and p53 are well-established oncoproteins and tumor suppressor proteins, respectively, with mechanisms of action that are distinct from those of the apoptosis-related proteins. The authors should carefully elaborate on how these proteins interact with additional modulators, such as p21, NOXA, PUMA, and MDM2, to clarify their role in the observed effects.

Moreover, to substantiate the proposed mechanism of action, it would be valuable to assess whether cell death induced by the Ribitol and Shikonin combination is attenuated upon temporary inhibition of key genes such as c-Myc, p53, Mcl-1, or Bcl-Xl, using RNA interference (RNAi) or gene knockdown/knock-out approaches. This would provide further insights into the causal relationship between these molecular targets and the observed phenotypic outcomes.

Lastly, since Figures 4B and 4C present the same data, I recommend consolidating these figures and incorporating the aforementioned mechanistic details into Figure 4C for clarity and conciseness.

Response: We appreciate the reviewer’ comments. We agree with the reviewer that cell death/survival related oncoproteins and tumor suppressor proteins are a complex signaling network, and their interactions control cancer cells death and growth. The 4 gene products we have investigated are arguably the key components of this network. We understand the importance of establishing a mechanism for the observed synergy of ribitol and shikonin, but we also realize the complexity of the network. In agreement with the reviewer, experiments are underway to exploring more signaling molecules within the network in cell culture and in vivo with the aim to establish more detailed mechanism of action. We therefore consider these further studies are beyond the current manuscript. Nevertheless, we follow the reviewer’s advice and modified the mechanistic explanation in now Figure 7 to illustrate the potential importance of the alteration in expression of these genes and in metabolism with the two drugs to their synergistic effect.  We also modified the explanation for the role the alteration in C-Myc and P53 expression to the observed drug effect in Discussion. Hope this will provide a much clear mechanistic view to leaders.   

In addition, we now provide a data of migration assay performed under Live-Cell imaging system showing the anti-metastatic potential of ribitol and shikonin combination (Figure 3C) (highlighted in text).

  1. Comment:

Figure 5:
As previously highlighted, the authors emphasize the synergistic anticancer effects of the Ribitol and Shikonin combination. However, to substantiate this hypothesis, the current presentation of Figure 5 has several notable shortcomings. Specifically, the analysis of cellular metabolic proteins related to the glycolysis or TCA cycle in the context of Ribitol and Shikonin treatment—when compared to treatments with Ribitol alone or Shikonin alone—introduces several inconsistencies and contradictions in the data interpretation.

To properly validate the proposed mechanism of synergy, the experimental design must be expanded to include a combination treatment group. In particular, if the authors wish to claim that cell death induced by the Ribitol and Shikonin combination is linked to mitochondrial-related metabolic alterations, this assertion must be rigorously demonstrated by including controls and all relevant treatment groups (control, Ribitol, Shikonin, and the combination of Ribitol and Shikonin). A more comprehensive analysis will provide clarity and strengthen the credibility of the hypothesis.

The authors should exercise particular caution when introducing novel anticancer candidates and discussing their effects in a manner that may appear superficial or inadequately substantiated. Such practices not only undermine the quality of the research but also pose potential risks to the integrity of global collaborative research endeavors. Rigorous experimental validation and a more detailed, transparent presentation of the data are essential for maintaining scientific credibility.

Response: We appreciate the reviewer’s comments. We agree that a combination treatment group is generally recommended for study to support the proposed mechanism of synergy. However, in our specific study condition, the combination of treatment leaded to early and effective cell death with very limited cells remaining. So far we found it very difficult to rely on data obtained from these remaining cells to make meaningful comparison. Using such data to make comparison and conclusion may therefore cause confusion and be misleading for mechanism. Further from application point of view, the selective nature of the synergy in cancer cell types suggests that the application of ribitol in cancer treatment will require identification of specific genotype and phenotype and response pattern of individual cancer to shikonin treatment, whereas the consequence of the combined treatment to the expression pattern of these genes would be less critical.

 We agree with the reviewer that caution should be taken when introducing novel anticancer candidates. We therefore added a short discussion as follow: “It should be noted that while ribitol is a metabolite in nature and safe in animal models and phase III clinical trials, the synergistic anti-cancer effect of ribitol is highly selective in partner drugs and in target cells. Furthermore, such effect clearly requires validation by in vivo animal model experiments. A much-detailed mechanistic exploitation is then warranted.” 

  1. Comment:

Figure 6: The rationale underlying the schematic model presented in Figure 6 remains unclear. The author does not provide sufficient justification for how the relationships between key proteins, such as c-Myc, p53, Mcl-1, and Bcl-Xl—established through experiments in Figure 4—are incorporated into this model. As it stands, this figure appears to be a speculative abstract hypothesis rather than a data-driven representation of the experimental findings.

There is a lack of consistency between the experimental content presented in the manuscript and the schematic model proposed in Figure 6. If the author intends to emphasize that Ribitol and Shikonin exhibit distinct mechanisms of action, which may contribute to their observed synergistic effects, this should be more clearly articulated. Furthermore, the author should consider the clinical implications of these candidate anticancer agents. Has the potential for side effects been adequately evaluated, particularly in the context of their application to breast cancer patients? It is crucial for the author to maintain a patient-centered perspective, particularly when proposing new therapeutic candidates.

In summary, the schematic model in Figure 6 should be directly informed by the experimental data presented in the manuscript. It is essential that the model be grounded in the findings thus far, and the author must clarify the rationale behind its construction. I urge the authors to revise this figure to better reflect the empirical evidence and to provide a more coherent and scientifically justified explanation.

Response: We apologize for causing confusion to the reviewer and would like to clarify the nature of figure 6. Indeed, we intended to use the schematic representation to summarize the key findings of the study and link them to the widely accepted pathways concerned, thus providing readers an easier way to interpret the possible mechanistic significance.  We consider this way would be more meaningful than a simple drawing of our hypothesis.  Following the reviewer advice, we have now modified the figure 6 (Now became figure 7) for better understanding of our findings. We are providing summary of whole data in figure 7B and modified text and legend accordingly. We are also showing the hypothetical model of how the changes of the key proteins might interact with metabolic pathways and playing the synergistic role with ribitol and shikonin.

We have now modified the legend as follow:

Figure 7: Metabolomics of MCF-7 cells treated with ribitol and shikonin. Key metabolites alterations of each pathway of central carbon metabolism showing in a schematic way. (A) Shikonin treatments down-regulated key intermediates of glycolysis (3-PG, pyruvate and lactate), TCA cycle (Citrate, α-KG, succinate, fumarate and malate), PPP pathway (Ribose-5-p) and nucleotide metabolites (IMP and AMP) levels. Shikonin also down-regulated the levels of glutamate and as well as glycine and serine. Ribitol particularly down-regulated the 3-phosphoglycerate, serine, end-product of glycolysis lactate, and TCA cycle intermediates succinate and malate. (B) Overall view of key metabolite alterations of MCF-7 cells treated with ribitol and shikonin within the established pathways. Red colored arrows represent changes with Shikonin treatment and green colored arrows represent changes with ribitol treatment. Down-regulated metabolites are indicated by down-point arrows, and up-regulated metabolites by upper-pointed arrows. Ribitol and shikonin combination regulates metabolomics, inhibits proliferation as well as migration of cells and induce apoptosis by targeting key enzymes of glycolysis.

Minor Review

Question 1: Introduction Section

In the 'Introduction' section, the authors describe that the combination of Ribitol and JQ1 demonstrated a prominent synergistic effect in the MDA-MB-231 breast cancer cell line, but no such effect was observed in the hormone-positive breast cancer cell lines, MCF-7 and T47D. If the current study claims that the combination of Ribitol and Shikonin had a similar effect across all cell lines, the authors must provide an explanation for this apparent discrepancy. Specifically, if the observed differences are due to a hormone-dependent event, this should be clearly discussed in the 'Discussion' section. The authors should incorporate relevant literature to support this explanation, as the introduction has already alluded to such a possibility. A more comprehensive clarification of the underlying mechanisms contributing to the observed differential effects would strengthen the manuscript.

Response: We appreciate the reviewer’s comments. Yes, now we discussed cell type specific response in discussion section stated as “In general, different cell types respond differently to different drugs based on their unique genotype and phenotype profiles and different cell populations within a tumor mass can exhibit varying responses to the same drug due to differences in microenvironment and gene expression profiles. Thus, distinct molecular signatures are crucial in mediating cell response to treatments and in defining drug targets. Earlier studies suggests that this principle is also applicable to antimetabolic drugs, especially in combination treatment. Heterogeneity in metabolism reported in broad range of cancers and specifically in different breast cancers by a multi-omics analysis indicates likelihood of differential response when antimetabolic drugs and metabolite intervention are considered as treatments [6,26-31].   

Question 2: Reference Citation

On page 9, line 203, there is still the word 'ref' that seems unnecessary.  Such mistakes cast doubt on the author's attention to detail and commitment to scholarly rigor. This should be rectified promptly. Furthermore, when making claims, particularly in the 'Introduction' and 'Discussion' sections, it is essential to include at least 2-5 references to substantiate the content. References are critical for bolstering the scientific credibility and impact of the work. Without proper citation, the manuscript may lose its trustworthiness. The authors should ensure that all necessary references are included to appropriately contextualize their findings and support the claims made throughout the manuscript.

Response: We apologize for this oversight and thank you for the observation. Now, we carefully checked to avoid those typo mistakes. As per reviewer suggestion, we added more references related to the context in introduction and discussion sections to get the essence of the flavor.

Reviewer 2 Report

Comments and Suggestions for Authors

In this article, the authors investigated the synergistic effect of ribitol and shikonin on breast cancer cells and shown that combination of both the compounds resulted in decreased cell proliferation and an increase in apoptotic cell populations. Results were interesting; however, the article can be improved if the following issues were addressed

1) In page 9 line 214, authors stated that "Further more, shikonin treatment decreased the levels of many amino acids, particularly cysteine, 214 histidine, valine, threonine, and isoleucine with statistical significance (Figure 5B)." However in the figure 5B, under shikonin treatment most of the reported amino acids were upregulated. Please clarify it.

2) Authors had shown which metabolites or amino acids were altered when treated alone with either of ribitol or shikonin. Can we expect to see similar changes in the levels of metabolites or amino acids, when the cells were treated with both compounds? Please elaborate on it, if not I would request authors to perform similar experiment for the combination treatment group.

Reviewer 3 Report

Comments and Suggestions for Authors

The objective of this study is to screen a panel of small molecules targeting energy metabolism to identify potential candidates in combination with Ribitol, which could exert synergistic inhibitory effects on breast cancer cells in vitro. The authors have reported that the combination of Ribitol and Shikonin has exerted synergistic inhibitory effects on three different breast cancer cell lines. Although the combination of treatments alone showed limited effects to reach efficacy, this manuscript addressed the combination indicating a potency in further pre-clinical study. However, this manuscript needs to address several important issues that would aid in the interpretation of the data.

Major comments

1. In Figure 1A and B, screened small molecules utilized various tested ranges for the selection of the most potential candidate in combination with ribitol. Could the authors address the reasons for determining the dose range of each molecule? (Refer to previous studies, or consider the physically achievable dose?)

2. Three different breast cancer cell lines were tested for the synergistic effect of the combination, and the author claimed the combination of shikonin with ribitol exhibited enhanced inhibitory effects over shikonin treatment alone in all three cell lines (line 114, page 5), however, only MCF7 was investigated for further profiling. Understanding the difference in response to the shikonin and ribitol combination across multiple cell lines will provide valuable knowledge for future studies. If the authors have performed studies on T-47D or MDA-MB-231, please include the data from their findings.

3. In Figure 4B, the authors proposed the combination treatment modulates the expression of apoptotic-related pathways (p53, Bcl-xL etc.) to explain the mechanism of action. However, the mechanism identified by this study may not be specific to the shikonin and ribitol combination treatment. Could the author pay more effort to metabolic-related pathways, for example, glycolysis/PPP pathway as the components in the combination mainly disturb tumor metabolism (PKM2, PPP pathway)?

4. Metabolomic analysis showed shikonin treatment impairs glycolysis which is likely a resistant mechanism induced by ribitol treatment alone. Could the authors confirm this finding by determining the glycolysis rate and extracellular acidification rate in MCF7 cells using seahorse assay?

Minor comments

1. Could the authors cite related references when claim the effective dose of shikonin in MCF7 cell viability study was lower than the concentration of physiologically relevant dose in vivo? (#Line 93, Page 2)

2. In line 98, page 3, the author showed shikonin EC50 is 11.43 ± 0.13 μM for T-47D cell line, but listed a different EC50 value (10.4 µM) in Figure 1.C (Page 4).

3. The authors showed the combination of shikonin and ribitol exerted synergistic inhibitory effects on cell viability using ATP-based measurement in Figure 2. Could the authors include representative images of in vitro cultured cells in response to the combination of shikonin (3µM) and ribitol (5mM) with the highest synergistic score to show the inhibitory effects on cell growth in Figure 3?

4. In Figure 4A, the authors showed the combination-induced MCF7 cell apoptosis at 48h and 72h by measuring the luminescence intensity of annexin-v/PI co-staining. Could the authors show representative immuno-fluorescent images of apoptotic cells when treated with the combination at 48h?

5. Could please re-organize and label the raw images of western blots related to Figure 4B? At least three repeats of WB are required.

6. Please indicate how many repeats were performed on metabolomics analysis in Figure 5 legend and show statistic marks in Figures 5A and B. In lines 214 and 215 on page 9, the authors claimed that shikonin decreased the levels of many amino acids, whereas fold change of most amino acids smaller than 0.5, please address this in both result the section.

7. Please proofread the manuscript for syntax errors.

Comments on the Quality of English Language

Moderate English changes required.

Round 2

Reviewer 1 Report

Comments and Suggestions for Authors

As a reviewer of this research paper, I would like to recommend the paper for publication. I believe that the authors have diligently addressed the questions and requests raised during the review process. While some aspects of the responses did not fully satisfy every point raised, I do not consider this as grounds for rejection. The authors have conducted thorough and meaningful research on the anticancer treatment, and their work demonstrates significant contributions to the field. I am confident that the revisions made have strengthened the manuscript, and I support its publication.

Comments on the Quality of English Language

The authors' writing skills require careful attention and further refinement. However, I believe that the overall process and findings of the paper have been conveyed effectively. Given the importance of clear communication in scientific writing, I would recommend that the journal review this aspect more thoroughly to ensure the clarity and coherence of the manuscript before publication.

Reviewer 2 Report

Comments and Suggestions for Authors

Accept the manuscript in the current version 

Reviewer 3 Report

Comments and Suggestions for Authors

The authors have addressed most of the concerns and the manuscript could be considered for publication.

Comments on the Quality of English Language

The written of current version is fine.